# The Validity and Reliability of the Microsoft Kinect for Measuring Trunk Compensation during Reaching

**DOI:** 10.3390/s20247073

**Published:** 2020-12-10

**Authors:** Matthew H. Foreman, Jack R. Engsberg

**Affiliations:** Program in Occupational Therapy, Washington University in St. Louis, 4444 Forest Park Ave., St. Louis, MO 63108, USA; engsbergj@wusm.wustl.edu

**Keywords:** trunk, upper extremity, compensation, reaching, Kinect, video motion capture, validity, reliability

## Abstract

Compensatory movements at the trunk are commonly utilized during reaching by persons with motor impairments due to neurological injury such as stroke. Recent low-cost motion sensors may be able to measure trunk compensation, but their validity and reliability for this application are unknown. The purpose of this study was to compare the first (K1) and second (K2) generations of the Microsoft Kinect to a video motion capture system (VMC) for measuring trunk compensation during reaching. Healthy participants (n = 5) performed reaching movements designed to simulate trunk compensation in three different directions and on two different days while being measured by all three sensors simultaneously. Kinematic variables related to reaching range of motion (ROM), planar reach distance, trunk flexion and lateral flexion, shoulder flexion and lateral flexion, and elbow flexion were calculated. Validity and reliability were analyzed using repeated-measures ANOVA, paired *t*-tests, Pearson’s correlations, and Bland-Altman limits of agreement. Results show that the K2 was closer in magnitude to the VMC, more valid, and more reliable for measuring trunk flexion and lateral flexion during extended reaches than the K1. Both sensors were highly valid and reliable for reaching ROM, planar reach distance, and elbow flexion for all conditions. Results for shoulder flexion and abduction were mixed. The K2 was more valid and reliable for measuring trunk compensation during reaching and therefore might be prioritized for future development applications. Future analyses should include a more heterogeneous clinical population such as persons with chronic hemiparetic stroke.

## 1. Introduction

Upper extremity (UE) motor impairments are highly prevalent in many clinical populations such as stroke [1]. Impaired UE movement is frequently accompanied by compensatory strategies that help a person adapt to limitations in motor function but may impact recovery and cause negative effects if used long term [2,3,4]. There are numerous well-researched, standardized assessments that measure UE abilities according to factors such as speed, strength, range of motion (ROM), and movement quality, but few that directly measure the amount of compensation utilized during task performance [5,6,7]. Without objective measurement and subsequent intervention, continued compensatory movements can reduce the amount of task-driven neuroplastic change achieved following neurologic injury and ultimately contribute to maladaptive plasticity, learned disuse or non-use, and chronic pain or injury [2,3,4]. Objective assessment of targeted and compensatory UE movements often relies on video motion capture cameras (VMC) or electromagnetic sensors that, while extremely accurate, are typically expensive and not feasible for application in a clinical setting. Because the amount of motor recovery achieved, and inversely the amount of compensation used, is highly predictive of participation and quality of life in persons living with long-term UE impairments, a clinically feasible, affordable, accurate, and objective measure of movement compensation may be an important innovation in rehabilitation science [8].

The Microsoft Kinect (Microsoft Corp., Redmond, WA, USA) is a low-cost, off-the-shelf motion sensor originally designed for video games that can be adapted for quantitative assessment of UE clinical movements [9,10,11,12]. The measurement abilities of the first-generation Kinect (K1) have been established for UE movements, spatiotemporal gait variables, standing balance, postural control, and even static foot posture [9,10,13,14,15]. The abilities of the second-generation Kinect (K2) are not as robustly established, but have been investigated for some UE, gait, and postural movements [11,16,17]. A recent study within our laboratory found both sensors to be valid relative to the gold standard of a VMC system when measuring reaching (forward and side) and angular shoulder movements (frontal, transverse, sagittal) [12]. Both sensors have also been frequently used within our laboratory for virtual reality (VR)-based motor rehabilitation aimed at improving UE motor abilities of persons with various impairments [18,19,20,21]. The Kinect sensors have some advantages over widely used optical and inertial sensor systems, namely significantly lower cost, higher portability, easier deployment in a lab or clinic, wider accessibility, and marker-less motion tracking with simpler throughput for the control of video games and VR applications. Conversely, the Kinects typically produce significantly lower resolution and less reliable data compared to gold-standard motion capture systems such as VMC and wearable inertial sensors [9,10,11,16].

Reaching is one of the most rigorously researched UE movements due to its involvement in many activities of daily living (ADLs). The kinematics of reaching in populations such as chronic stroke have been investigated in many different studies that often rely on VMC systems [22,23]. Not only do persons with stroke reach less accurately, slower, and with less motor control, they also utilize trunk flexion earlier and to a greater degree compared to the healthy population [22]. While differences in symmetry and joint coordination exist between healthy and impaired reaching, placing objects beyond the arm’s length of healthy participants has been found to elicit trunk movement similar to that used by hemiparetic stroke patients reaching to objects within arm’s length [23]. Few previous studies have examined the abilities of both generations of the Kinect sensor for measuring trunk compensation during reaching [24], and only one existing study has compared the measurement abilities of both sensors to simultaneous video motion capture [12]. The current study aims to go beyond previous work performed in our laboratory [12] to include a larger sample size of participants and movement trials with a focus on trunk kinematics during reaches that require trunk compensation. The purpose of this investigation was to establish the validity and reliability of two versions of the Microsoft Kinect for measuring UE and trunk kinematics during different reaching conditions.

## 2. Materials and Methods

### 2.1. Participants

A convenience sample of five healthy participants (3 women and 2 men, mean age 24.8 years) were recruited to participate in this study. A small sample size was considered due to the large sample of reaches (240 repetitions) performed by each participant and the overall focus of this study being the comparison of repeatable reaching motions across sensors and testing days. All participants gave informed written consent and the study protocol was approved by the university’s Institutional Review Board.

### 2.2. Hardware

Both the K1 and K2 combine standard red-green-blue (RGB) video and an infrared (IR) depth sensor with advanced pattern recognition algorithms to provide full-body, three-dimensional (3D) skeletal motion capture without the use of wearable trackers. Both sensors provide data at approximately 30 frames per second (fps), but the K2 generally boasts improved hardware compared to the K1 (Table A1) [25]. For example, the K2 collects high definition RGB images (1920 × 1080 pixels) while the K1 collects standard definition RGB (640 × 480 pixels) that fails to compete with most modern webcams [25]. The RGB and IR cameras in the K2 also have wider fields of view and, when combined with updated tracking algorithms, can track greater numbers of skeletal landmarks and overall users [25]. Most importantly, the K2 utilizes a time-of-flight algorithm for motion tracking that is more robust, less noisy, and more reliable than the structured light algorithm used by the K1 [25]. The VMC system was considered the gold standard for comparison in this case and consisted of eight IR motion capture cameras (MAC Eagle Digital Cameras, Motion Analysis Corp., Santa Rosa, CA, USA) measuring at 60 fps with a 3D resolution accurate to within one millimeter.

### 2.3. Experimental Procedure

Participants performed a set of targeted reaching movements similar to a previously developed reaching performance task [12,26] while simultaneously being measured by the K1, the K2, and an 8-camera VMC system. Each participant was seated on a stool in the center of the VMC capture volume with the K1 and K2 positioned at a midline distance of approximately 2.0 m and a height of 1.2 m [12]. Each movement set involved reaching towards a target in the sagittal (forward), scaption (45 degree angle), or frontal (lateral) planes at either a non-extended or extended distance. The non-extended distance was defined relative to each participant’s anthropometrics as shoulder height and arm’s length, while the extended distance was moved 20 cm beyond arm’s length (Figure 1). This extended reach required a healthy participant to flex the trunk and displace the shoulder to meet the target, similar to compensatory movements employed for reaching by persons with hemiparetic stroke [23]. Participants were provided verbal instruction but, given that they were healthy participants performing a relatively simple targeted reaching movement, no formal training was provided. On two different testing days, five repetitions were performed within each of four sets for the three directions and two conditions, resulting in a total of 240 repetitions for each of five participants. Given the large number of movements, participants were consistently asked for signs of fatigue and pain. None of the healthy participants reported any pain or fatigue in the UE. Participants were also given short breaks between movement sets (approx. 3–5 min) to mitigate fatigue. These breaks allowed researchers to code and save data files, check for data errors, and double check or adjust experimental setup and procedures.

### 2.4. Data Collection

Kinematic data were collected for the K1 and K2 using the Microsoft Kinect for Windows Software Development Kit (SDK v1.8 and v2.0) [27], a virtual reality peripheral network (VRPN) server [28], and custom software designed in MATLAB (r2012a, Mathworks Inc., Natick, MA, USA). The 3D positions of 11 upper body landmarks for the K1 and K2 were measured relative to each sensor’s origin (Figure 2). Common landmarks were head, neck, shoulders, elbows, wrists, and hands. The K1 defined torso as the body centroid, while the K2 defined the torso as a mid-spine landmark. Similar data were simultaneously collected for the VMC system using Motion Analysis software (Cortex, Motion Analysis Corp., Santa Rosa, CA, USA) to measure the positions of 25 retroreflective markers placed on bony landmarks on the participant’s upper body. Markers were placed on the top of the head (vertex); C7, T10, L5, and S4 vertebrae; sternal notch; xiphoid process; acromion processes; medial and lateral epicondyles; ulnar and radial styloids; anterior superior iliac spines; dorsal hands; and index fingers. Two redundant markers were placed on the humerus and forearm.

### 2.5. Analysis Procedure

Once collected, Kinect data were filtered (6th order, 6 Hz Butterworth) and used to create body segment vectors including spine (torso-neck), humerus (shoulder-elbow), and forearm (elbow-wrist/hand). VMC data were similarly filtered (6th order, 6 Hz Butterworth), imported into MATLAB, and used to create analogous body segments using marker midpoints and biomechanical conventions [29]. Clinically relevant variables were calculated including reaching ROM, planar reaching distance (sagittal and frontal), shoulder flexion and abduction, trunk flexion and lateral flexion, and elbow flexion. Reaching ROM was defined as the Euclidean distance between the shoulder and the hand, while planar reaching distance was defined as the distance traveled by the hand in the sagittal or frontal plane. Shoulder flexion and abduction were defined as the angle between the humerus and spine in the sagittal and frontal planes, respectively. Trunk flexion and lateral flexion were similarly defined as the angle between the spine and the vertical coordinate axis in the sagittal and frontal planes, respectively. Finally, elbow flexion was defined as the angle between the forearm and the humerus.

### 2.6. Statistical Approach

A peak detection algorithm was used to determine the start and stop of each reach in terms of the maximum and minimum distance of the hand from the target. The target’s position was not inherently available from the Kinect data, therefore an estimation was calculated as the average hand position at its maximum Euclidean distance from neutral. The first repetition of each trial was disregarded due to variable starting positions of the arm and hand. A three standard deviation algorithm was used to identify and remove outliers due to motion tracking errors. Validity was investigated using data from the first testing day (D1) to calculate magnitude differences, Pearson’s correlations (r), Bland-Altman 95% limits of agreement (LOA), and a repeated measures analysis of variance (ANOVA) with Bonferroni corrections across sensors. Reliability was investigated using averages within each testing day to calculate magnitude differences, intra-class correlations (ICC), Pearson’s correlations (r), Bland-Altman 95% LOA, and paired *t*-tests between days [30,31]. Estimates of correlations in terms of r and ICC were evaluated as excellent (0.75–1), modest (0.4–0.74), or poor (0–0.39) [31]. Bland-Altman analyses for validity (Table A2) and reliability (Table A4) as well as Pearson’s correlations for reliability (Table A3) are presented in the Appendix A.

## 3. Results

### 3.1. Trunk Compensation

For trunk flexion and trunk lateral flexion, the K2 was closer in magnitude to the VMC than the K1 in all directions and for both non-extended and extended reaches (Table 1). For trunk flexion, when considering Bland-Altman LOA for all movements, the K2 was within −3.5°–6.6° and the K1 was within −2.7°–14.2° of the VMC (Table A2). Similarly for trunk lateral flexion, the K2 was within −5.9–7.9° and the K1 was within −9.0–13.4° of the VMC. Significant differences were found between K2 and VMC for trunk flexion during extended forward reaching and lateral flexion during extended scaption reaching (Table 1). Significant differences were found between K1 and VMC for trunk flexion during all extended reaches and lateral flexion in all conditions but extended lateral reaching.

The K2 was more valid than the K1 for measuring trunk movements during extended reaches (Table 2). The K2 showed excellent agreement with the VMC for measuring trunk flexion (r = 0.77–0.88) and lateral flexion (r = 0.77–0.89) during extended reaches. The K1 showed moderate–excellent agreement with the VMC for trunk flexion (r = 0.52–0.78) and moderate agreement for lateral flexion (r = 0.50–0.60) during extended reaches. For non-extended reaches, the K2 showed only moderate agreement (r = 0.43) for measuring trunk flexion during lateral reaching. All other correlations were poor for both the K1 and K2. Bland-Altman analyses show that mean biases for trunk flexion and lateral flexion were smaller and with narrower LOA for the K2 than the K1 when compared to VMC (Table A2).

Reliability results were mixed for all three sensors when measuring the trunk (Table 3). The K2 showed excellent reliability for measuring trunk flexion during lateral reaching (ICC = 0.91), but poor–modest reliability for trunk flexion in all other reach directions (ICC = −0.53–0.69). The K2 also showed excellent reliability for lateral flexion in the scaption (ICC = 0.75), lateral (ICC = 0.82), and extended forward (ICC = 0.84) directions, but poor–modest reliability in all other directions (ICC = 0.12–0.66). The K1 showed modest–excellent reliability (ICC = 0.62–0.88) for trunk measurements during reaches in all directions except forward (ICC = 0.28–0.34). The VMC showed mixed results similar to K2, with poor–excellent reliability in the forward direction (ICC = −0.42–0.93), poor–excellent reliability in the scaption direction (ICC = 0.08–0.89), and modest–excellent reliability in the lateral direction (ICC = 0.66–0.82) for both trunk flexion and lateral flexion. Pearson’s correlations between testing days mirror these results (Table A3). Bland-Altman LOA analyses show small mean biases for trunk flexion and lateral flexion between testing days for the K1 (bias = −1.4°–0.8°), K2 (bias = −3.2°–1.4°), and VMC (bias = −3.0°–1.8°) (Table A4).

### 3.2. Upper Extremity Movements

The movement traces for the three planar reaching conditions (i.e., sagittal, scaption, frontal) illustrate directional differences between the Kinects and the VMC (Figure 3). Discrepancies in reaching magnitude between the Kinects and the VMC were dependent on the direction of movement. Differences in reaching ROM and planar distance were greatest during forward reaching, reduced during scaption reaching, and least during lateral reaching (Figure 3). Reaching ROM, planar reach distance, and elbow flexion measurements consistently showed excellent validity for the K2 (r = 0.79–0.99) and moderate–excellent validity for the K1 (r = 0.60–0.95) (Table 2). Reliability of these measurements was moderate–excellent for all three sensors (Table 3). Validity and reliability of shoulder flexion and abduction measurements varied from poor to excellent for all three sensors (Table 2 and Table 3).

## 4. Discussion

The purpose of this investigation was to establish the validity and reliability of two versions of the Microsoft Kinect for measuring UE and trunk kinematics during various reaching conditions. Specifically, participants were asked to perform both a non-extended and extended reach in each of three directions (forward, scaption, lateral) while their movements were recorded by the K1, K2, and the gold-standard VMC simultaneously. The K2 measured the trunk more similarly to the VMC as shown by smaller average magnitude differences in trunk flexion and lateral flexion. Validity results for trunk measurement were excellent for the K2 and modest–excellent for the K1 during extended reaching conditions intended to simulate movements that might be used by persons with chronic stroke. Reliability for trunk measurement was modest–excellent for extended reaching with the K1, with the exception of the forward direction, but varied from poor to excellent for the K2. Results for both sensors were generally excellent for measuring arm and hand displacement, excellent for measuring elbow flexion, and mixed for shoulder measurement, with reaches in the scaption and lateral directions providing more valid and reliable results than the forward direction.

The results of this study are supported by previous research that examines the validity of the K1 and K2 in terms of other functional movements. Bonnechere and colleagues [9] found similar results when comparing the K1 to VMC during the performance of four functional movements including shoulder abduction (similar to lateral reaching) and elbow flexion (similar to forward reaching). Clark and colleagues [11] found the K2 to have excellent concurrent validity for measuring trunk movements during dynamic balance tasks and anterior–posterior movements, but poor–moderate validity for static tasks and medial–lateral movements. In the current investigation, the K2 similarly shows the greatest validity for measuring trunk flexion during an extended movement in the anterior–posterior direction. Reither et al. [12] found similar results while measuring the K1, K2, and VMC simultaneously with a single participant reaching forward, reaching to the side, and performing shoulder movements in various planes, but did not investigate trunk kinematics during such movements. In summary, Reither et al. [12] similarly found a greater range in single-day correlations between K1 and VMC (r = 0.31–0.96) than between the K2 and VMC (r = 0.45–0.96) with correlation magnitudes dependent on movement plane. The authors also found varied day-to-day reliability results for both K1 and K2 and, in general, a greater direction-dependent underestimation of kinematics displayed by the K1 [12]. The current study goes beyond the methods of Reither et al. [12] by utilizing an increased sample size of participants and movements, the inclusion of extended reaches to elicit trunk compensations, analysis of the trunk along with the UE, and movements in the scaption plane along with sagittal, frontal, and transverse planes.

We found several low and negative reliability (ICC) values (Table 3), particularly for shoulder flexion, shoulder abduction, trunk flexion, and trunk lateral flexion during non-extended reaching in the forward and scaption directions for all sensors including VMC. Negative ICC values are not ideal and can often be attributed to low between-subjects variance in the phenomenon being measured [32]. Accordingly, these results might be due to small between-day variance in the kinematic variables being tested. For example, a negative ICC value (ICC = −0.53) was calculated for the K2 between days for trunk flexion during the extended forward reach, but Bland-Altman analysis shows a small mean bias (bias = −3.0°) and LOA (LOA = −13.2–6.8°). This suggests a relatively small mean difference, and thus satisfactory repeatability, between testing days even in the face of a negative ICC calculation that may be due to small and non-systematic variance. A more heterogeneous clinical population may improve correlation results by increasing variance in the sample. Pearson’s correlations (Table A3) and Bland-Altman LOA (Table A4) were included to give a broader picture of absolute and relative reliability for all three sensors. Additional, more advanced analyses may also provide further insight into these discrepancies; for example, dynamic time warping (DTW) is an advanced signal processing technique that could provide a measure of signal match for the time series data collected by the K1 and K2 [33].

The most notable limitation to this work is the use of healthy participants rather than a sample of participants with hemiparesis. As mentioned previously, persons with hemiparesis reach significantly differently than unimpaired persons, namely with slower movement, less accuracy, impaired interjoint coordination, and increased use of compensatory movement at the trunk [22,23]. Targets placed beyond the reach of healthy participants can elicit a similar compensatory response at the trunk, but persons with hemiparesis exhibit less symmetry and earlier trunk recruitment in comparison [23]. Healthy reaching is simply not the same as hemiparetic reaching. However, the purpose of the current study is to validate the measurement capabilities of the K1 and K2 relative to each other and to a gold-standard VMC system. Numerous referenced studies use healthy participants for sensor validation with intentions for future clinical application [9,10,11,12,13,14,15,16,17]. Healthy participants are more accessible, can perform the large number of required movements without fatigue or pain, and can more readily reproduce movements across trials and testing days for validity and reliability analyses. Given that the ultimate application of this study is implementation for clinical measurement of neurologically impaired populations, the ecological validity of future work would greatly benefit from testing with a more heterogeneous sample of persons with hemiparetic stroke.

The current study provides some insights for the design of such future work; for example, it may be necessary to recruit more individuals and reduce the overall repetitions performed to better capture variability, mitigate fatigue, and enhance the generalizability of results for real-world clinical populations. In addition, the experimental protocol could be adjusted to provide detailed instruction and training for impaired populations to reduce trial variability and enhance the efficiency of data reduction and cleaning. Given that the evidence shows that persons with hemiparetic stroke recruit the trunk earlier and more often than healthy populations [23], it may be necessary to eliminate or reduce the distance of the extended reach to maximize reaching performance and reduce frustration. Finally, given the results of the current study, it may be prudent to focus on the planes of movement best measured by the K1 and K2 due to their hardware constraints (e.g., lateral > scaption > forward).

Other variations in results might be attributed to various study limitations. First, the Kinect SDK uses a tracking algorithm that does not rely on the specific placement of markers on palpable bony landmarks as does the VMC. While this is convenient for users, it has been previously noted as a limitation in the Kinect’s ability to accurately measure kinematics of movement due to variable body segment lengths; however, previous studies have developed algorithms through regression that may be able to correct for this during real-time tracking [9]. Second, it was clear through both observation and the relatively high standard deviations attributed to each movement (Table 1) that different strategies were used for reaching by individual participants. No neutral starting point was defined a priori, and some participants returned their arm to their lap between repetitions while others remained in a flexed position. This resulted in large variations in range of motion, namely with elbow flexion. Finally, reliability results varied inconsistently for all three sensors, and it should be noted that, on top of statistical limitations, there are intra-individual differences across trials and across days in each participant’s reaching kinematics. Participants were given similar instructions for each trial and testing day, but differences in the repeatability of human movement yet exist and may be attributable to the slight variance in between-day correlation and significance testing. Participants were provided verbal instruction but no formal training at the simple reaching movements, so movement may have differed between movement sets and even testing days due to subtle learning effects. It is also possible that the placement of motion capture markers varied slightly between days, resulting in reliability differences. Increasing the overall sample size in the future could mitigate these intra- and inter-individual differences in repeatable movement.

This study shows that the K1 and K2 may serve as useful tools for objectively measuring UE and trunk kinematics, but application may depend on the body segment, joint, and movement plane of interest. Few studies have investigated their relative measurement properties, but both sensors are widely employed as the basis for VR-based interventions for persons with motor impairments including stroke and cerebral palsy [19,21]. Use of such interventions continues to grow along with client interest, professional knowledge, and technological accessibility [34]. The current investigation may inform future VR development, namely the inclusion of real-time measurement of trunk compensation using the K2.

## 5. Conclusions

In conclusion, the K1 and K2 have been shown to be valid and reliable for measuring some aspects of UE and trunk kinematics during reaching. In particular, the K2 exhibited slightly better characteristics for measuring the trunk during standard and extended reaching in different directions, and may be recommended over the K1 in future development for purposes of measuring trunk compensation in clinical populations.

## Figures and Tables

**Figure 1 sensors-20-07073-f001:**
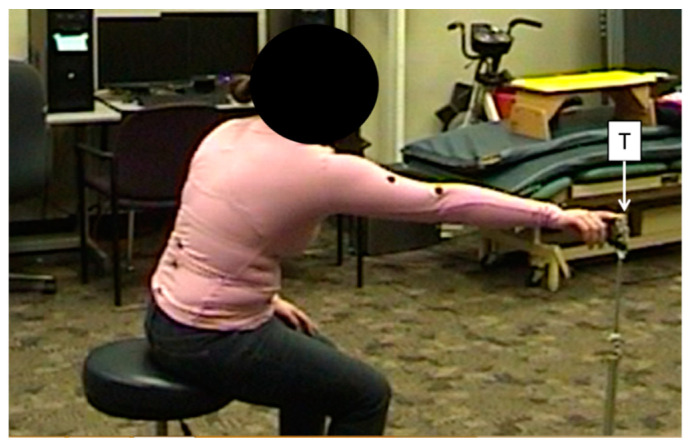
An example of a participant reaching towards the target (T) during an extended scaption reach while wearing retroreflective markers.

**Figure 2 sensors-20-07073-f002:**
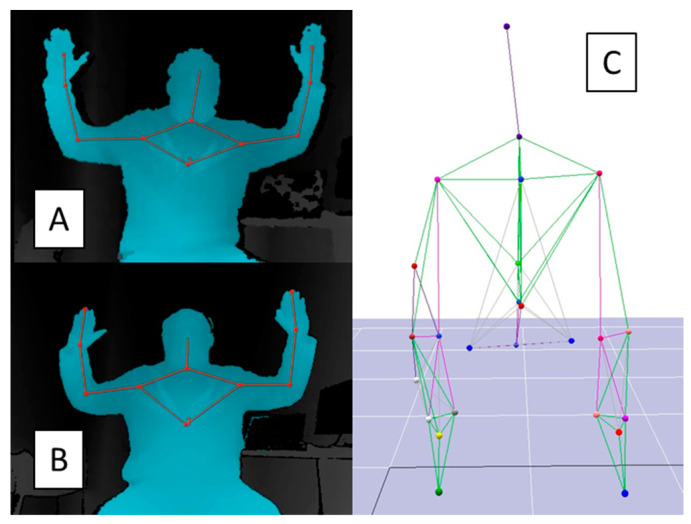
Examples of the kinematic body landmarks measured by the K1 (**A**), K2 (**B**), and VMC (**C**). The K1 and K2 measured 11 body landmarks. The VMC measured the position of 25 body landmarks.

**Figure 3 sensors-20-07073-f003:**
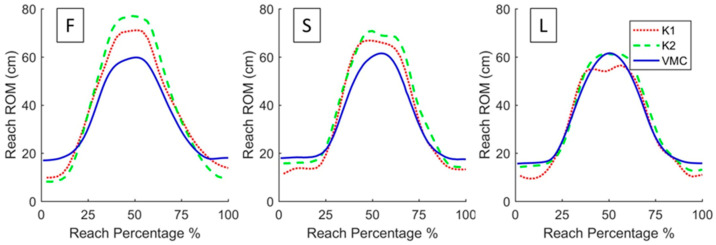
Three sets of curves showing reach ROM from start to stop of a typical reaching movement. The **left** curve (F) represents a forward reach, the **middle** curve (S) represents a scaption reach, and the **right** curve (L) represents a lateral reach. Curves for the K1, K2, and VMC are shown separately (see legend).

**Table 1 sensors-20-07073-t001:** Mean (± SD) magnitudes for the K1, K2, and VMC for all kinematic variables and all movements on the two different testing days D1 and D2. Each of five participants performed four sets of five reaches (N = 100) for each direction and condition. This sample was repeated on two separate testing days (D1 and D2).

		D1			D2		
		K1	K2	VMC	K1	K2	VMC
*Forward (N = 100)*							
	Reaching ROM (cm)	43.9 ± 11.6 *	49.2 ± 15.7 *	32.7 ± 14.0	36.3 ± 13.2	42.1 ± 19.6	25.4 ± 17.3
	Sagittal reach distance (cm)	49.2 ± 4.4	54.0 ± 12.2	42.0 ± 6.1	45.4 ± 6.5	49.5 ± 14.9	41.1 ± 6.6
	Shoulder flexion (deg)	77.7 ± 7.5 *	78.0 ± 6.5 *	60.8 ± 6.1	83.0 ± 10.7	74.7 ± 9.0	62.9 ± 5.3
	Trunk flexion (deg)	−2.2 ± 0.9	−0.4 ± 0.8	0.4 ± 1.3	−2.7 ± 1.2	−0.2 ± 0.5	0.2 ± 1.5
	Trunk lateral flexion (deg)	0.6 ± 0.5 *	0.0 ± 0.4	−1.1 ± 0.7	0.6 ± 0.5	−0.1 ± 0.4	−3.1 ± 11.3
	Elbow flexion (deg)	110.1 ± 43.6	104.4 ± 38.1	86.9 ± 29.5	87.5 ± 50.9	80.1 ± 48.3	69.0 ± 33.0
*Forward Extend (N = 100)*							
	Reaching ROM (cm)	37.4 ± 18.3	52.2 ± 19.4 *	30.3 ± 13.7	34.5 ± 16.5	51.5 ± 16.8	26.7 ± 15.4
	Sagittal reach distance (cm)	58.6 ± 12.8	72.7 ± 14.2	62.6 ± 6.8	57.7 ± 12.3	72.2 ± 15.9	61.0 ± 9.4
	Shoulder flexion (deg)	88.5 ± 9.5	103.0 ± 8.4 *	81.7 ± 7.2	87.8 ± 15.4	95.6 ± 16.3	78.8 ± 9.0
	Trunk flexion (deg)	10.3 ± 2.7 *	15.0 ± 3.5 *	18.7 ± 3.0	11.7 ± 1.6	17.9 ± 2.9	21.7 ± 3.5
	Trunk lateral flexion (deg)	0.9 ± 1.8 *	−0.9 ± 1.4	−3.7 ± 2.5	1.0 ± 2.1	−0.9 ± 2.3	−4.2 ± 5.7
	Elbow flexion (deg)	109.3 ± 44.7	111.3 ± 45.2	87.9 ± 29.3	99.4 ± 42.0	97.2 ± 43.0	73.9 ± 32.1
*Scaption (N = 100)*							
	Reaching ROM (cm)	39.1 ± 14.4	37.3 ± 16.7	33.3 ± 15.5	34.5 ± 13.4	30.6 ± 17.8	27.8 ± 17.3
	Sagittal reach distance (cm)	25.0 ± 5.9	26.8 ± 11.6	24.0 ± 6.0	25.9 ± 5.1	23.4 ± 11.6	24.0 ± 5.0
	Frontal reach distance (cm)	42.9 ± 7.5 *	45.2 ± 10.4	37.8 ± 6.5	37.9 ± 7.5	39.2 ± 10.2	34.3 ± 5.5
	Shoulder flexion (deg)	65.9 ± 12.3 *	57.7 ± 9.9 *	41.4 ± 11.6	67.4 ± 7.9	61.1 ± 6.5	46.3 ± 5.4
	Shoulder abduction (deg)	52.8 ± 17.2 *	55.2 ± 13.5 *	36.1 ± 11.4	57.7 ± 11.1	60.1 ± 9.9	37.0 ± 6.8
	Trunk flexion (deg)	−3.4 ± 1.0 *	−0.2 ± 0.7	0.0 ± 0.8	−3.7 ± 1.2	−0.2 ± 0.6	0.3 ± 1.1
	Trunk lateral flexion (deg)	−7.3 ± 1.5 *	−0.1 ± 0.5	−0.4 ± 0.8	−6.6 ± 1.8	−0.1 ± 0.5	−0.3 ± 0.9
	Elbow flexion (deg)	112.1 ± 44.7	101.1 ± 42.4	88.2 ± 33.3	88.5 ± 51.5	82.9 ± 41.6	74.4 ± 34.0
*Scaption Extend (N = 100)*							
	Reaching ROM (cm)	36.6 ± 14.6	40.5 ± 17.8 *	31.8 ± 14.5	28.3 ± 16.8	31.5 ± 19.3	24.7 ± 17.8
	Sagittal reach distance (cm)	30.4 ± 6.3	37.3 ± 11.2	37.8 ± 7.3	31.3 ± 6.5	34.3 ± 11.6	38.8 ± 5.3
	Frontal reach distance (cm)	47.4 ± 14.2	54.6 ± 14.9	51.2 ± 9.3	42.7 ± 11.0	48.6 ± 13.5	46.1 ± 7.4
	Shoulder flexion (deg)	72.0 ± 8.7	88.9 ± 7.6 *	62.1 ± 7.5	72.2 ± 8.1	87.8 ± 5.0	65.1 ± 6.9
	Shoulder abduction (deg)	66.7 ± 11.0 *	86.3 ± 8.9 *	58.5 ± 7.3	67.6 ± 8.6	87.6 ± 7.4	57.5 ± 7.2
	Trunk flexion (deg)	5.5 ± 1.4 *	12.5 ± 2.5	15.0 ± 2.8	6.4 ± 1.9	13.1 ± 2.5	15.1 ± 4.9
	Trunk lateral flexion (deg)	10.2 ± 1.7 *	13.4 ± 2.4 *	16.0 ± 3.5	10.8 ± 3.1	13.3 ± 3.5	15.9 ± 3.9
	Elbow flexion (deg)	107.4 ± 44.9	108.8 ± 42.7	88.6 ± 32.6	85.7 ± 48.6	86.4 ± 46.5	72.0 ± 36.6
*Lateral (N = 100)*							
	Reaching ROM (cm)	25.6 ± 16.0	27.7 ± 16.8	29.0 ± 16.7	23.3 ± 14.0	22.6 ± 16.5	27.4 ± 17.9
	Frontal hand distance (cm)	49.7 ± 10.8	57.8 ± 12.6	51.6 ± 5.4	44.6 ± 10.2 **	50.9 ± 14.9 **	47.1 ± 7.5
	Shoulder abduction (deg)	51.3 ± 12.2	53.1 ± 10.5 *	42.6 ± 10.3	49.3 ± 12.0	49.5 ± 12.7	39.2 ± 9.6
	Trunk flexion (deg)	0.3 ± 0.9	0.2 ± 0.7	−0.2 ± 0.7	0.4 ± 0.9	0.0 ± 0.3	0.1 ± 0.6
	Trunk lateral flexion (deg)	−7.8 ± 1.3 *	−0.6 ± 0.9	0.0 ± 1.4	−7.7 ± 2.5	−0.5 ± 0.6	−0.5 ± 0.9
	Elbow flexion (deg)	91.1 ± 48.4	91.6 ± 44.5	79.2 ± 35.9	84.8 ± 48.8	80.2 ± 42.9	72.9 ± 36.6
*Lateral Extend (N = 100)*							
	Reaching ROM (cm)	13.1 ± 14.8 *	23.7 ± 16.1	25.3 ± 15.4	13.8 ± 17.3	20.5 ± 18.2	24.5 ± 18.1
	Frontal hand distance (cm)	55.7 ± 13.1 *	69.9 ± 14.9	69.4 ± 7.6	52.6 ± 15.2	65.2 ± 19.6	65.5 ± 11.8
	Shoulder abduction (deg)	77.4 ± 8.5	88.5 ± 9.2 *	72.9 ± 9.4	72.1 ± 10.3	81.0 ± 11.4	67.4 ± 11.5
	Trunk flexion (deg)	0.0 ± 2.2 *	3.8 ± 3.0	3.8 ± 3.1	−0.8 ± 1.3	2.5 ± 2.7 **	3.0 ± 2.9
	Trunk lateral flexion (deg)	18.9 ± 3.9	21.7 ± 3.4	23.9 ± 4.6	18.4 ± 4.9	20.5 ± 3.8	22.1 ± 6.4
	Elbow flexion (deg)	87.5 ± 48.4	93.6 ± 42.6	77.3 ± 32.4	81.3 ± 51.6	86.9 ± 46.6	73.2 ± 36.3

** p* < 0.05 for Bonferonni-corrected pairwise *t*-test between Kinect and VMC. ** *p* < 0.05 for paired *t*-test between testing days. K1: KinectV1; K2: KinectV2; VMC: video motion capture; D1: day one of testing; D2: day two of testing.

**Table 2 sensors-20-07073-t002:** Validity measured by Pearson’s correlation coefficients (r) between the K1 and VMC and the K2 and VMC on D1.

		Forward		Scaption		Lateral	
		K1	K2	K1	K2	K1	K2
*Non-Extended*							
	Reaching ROM (cm)	0.93 *	0.95 *	0.94 *	0.94 *	0.94 *	0.94 *
	Sagittal reach distance (cm)	0.60 *	0.79 *	0.75 *	0.81 *	-	-
	Frontal reach distance (cm)	-	-	0.93 *	0.97 *	0.92 *	0.94 *
	Shoulder flexion (deg)	0.19	0.24	0.77 *	0.80 *	-	-
	Shoulder abduction (deg)	-	-	0.97 *	0.97 *	0.88 *	0.96 *
	Trunk flexion (deg)	−0.19	0.01	−0.44 *	0.22	−0.03	0.17
	Trunk lateral flexion (deg)	0.25 *	0.10	−0.36 *	0.20	−0.23 *	0.43 *
	Elbow flexion (deg)	0.95 *	0.94 *	0.97 *	0.99 *	0.96 *	0.99 *
*Extended*							
	Reaching ROM (cm)	0.95 *	0.91 *	0.90 *	0.98 *	0.90 *	0.95 *
	Sagittal reach distance (cm)	0.91 *	0.82 *	0.67 *	0.84 *	-	-
	Frontal reach distance (cm)	-	-	0.97 *	0.96 *	0.94 *	0.95 *
	Shoulder flexion (deg)	0.23 *	0.36 *	0.31 *	0.66 *	-	-
	Shoulder abduction (deg)	-	-	0.90 *	0.91 *	0.72 *	0.89 *
	Trunk flexion (deg)	0.78 *	0.88 *	0.52 *	0.77 *	0.72 *	0.83 *
	Trunk lateral flexion (deg)	0.51 *	0.77 *	0.60 *	0.89 *	0.50 *	0.78 *
	Elbow flexion (deg)	0.98 *	0.97 *	0.96 *	0.98 *	0.99 *	0.99 *

** p* < 0.05 for Pearson’s correlation between Kinect and VMC. K1: KinectV1; K2: KinectV2; VMC: video motion capture; D1: day one of testing.

**Table 3 sensors-20-07073-t003:** Reliability measured by intra-class correlation coefficients (ICC) between testing days D1 and D2 for each of the three sensors.

		Forward			Scaption			Lateral		
		K1	K2	VMC	K1	K2	VMC	K1	K2	VMC
*Non-Extended*										
	Reaching ROM (cm)	0.59	0.86	0.78	0.88	0.82	0.84	0.98	0.96	0.99
	Sagittal reach distance (cm)	0.54	0.90	0.82	0.58	0.73	0.74	-	-	-
	Frontal reach distance (cm)	-	-	-	0.74	0.77	0.76	0.93	0.94	0.84
	Shoulder flexion (deg)	0.37	0.46	−0.08	0.52	−0.22	−0.02	-	-	-
	Shoulder abduction (deg)	-	-	-	0.56	0.56	0.84	0.99	0.93	0.96
	Trunk flexion (deg)	0.28	0.31	0.93	0.74	0.43	0.89	0.86	0.69	0.66
	Trunk lateral flexion (deg)	0.34	0.14	0.84	0.69	0.75	0.36	0.85	0.82	0.81
	Elbow flexion (deg)	0.75	0.82	0.74	0.76	0.72	0.80	0.99	0.97	0.99
*Extended*										
	Reaching ROM (cm)	0.95	0.99	0.94	0.85	0.77	0.83	0.97	0.96	0.99
	Sagittal reach distance (cm)	0.90	0.98	0.79	0.65	0.67	0.78	-	-	-
	Frontal reach distance (cm)	-	-	-	0.92	0.87	0.70	0.97	0.97	0.92
	Shoulder flexion (deg)	−0.57	−0.23	−0.61	0.17	−0.69	−1.47	-	-	-
	Shoulder abduction (deg)	-	-	-	−0.18	−0.04	0.26	0.52	0.66	0.88
	Trunk flexion (deg)	0.74	−0.53	−0.42	0.75	0.66	0.74	0.63	0.91	0.82
	Trunk lateral flexion (deg)	0.87	0.84	0.65	0.62	0.20	0.08	0.88	0.66	0.71
	Elbow flexion (deg)	0.94	0.92	0.91	0.82	0.75	0.82	0.98	0.98	0.97

K1: Kinect V1; K2: KinectV2; VMC: video motion capture; D1: day one of testing; D2: day two of testing.

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
