# Peer review of "The Validity and Reliability of the Microsoft Kinect for Measuring Trunk Compensation during Reaching"

_sensors, 2020, doi:10.3390/s20247073_

Round 1

Reviewer 1 Report

The validity and reliability of the Microsoft Kinect 3 for measuring trunk compensation during reaching

This paper is a technical paper for measuring reliability and validity. This research is difficult to publish but must continue to be carried out in order to know the clinometric qualities of the tools. This type of studies allows to use or not to use the tools validated in clinical practice. I encourage the publication of this type of article.

Abstract is clear and well structured.

Introduction

The authors should explain why the second generation kinect might be likely to have different clinometric qualities. Better ??

Material

Why only 5 participants ?

Given the number of tests, isn't there a risk of fatigue and therefore a measurement bias?

Statistic is appropriate

Results

The kinematic curves could also have been analyzed with dynamic time warping (DTW) analysis

See DYSKIMOT: An ultra-low-cost inertial sensor to assess head’s rotational kinematics in adults during the Didren-Laser Test, Hage et al  in Sensors 2020

Author Response

Greetings,

I want to thank you for your thoughtful comments regarding my manuscript.  I have replied to each comment below and highlighted changes in the new uploaded manuscript.  I have attempted to address each of your comments.  Please let me know if you have any questions and I look forward to your further review.

Thanks again,

Matthew Foreman, PhD

Reviewer 1

The validity and reliability of the Microsoft Kinect 3 for measuring trunk compensation during reaching

This paper is a technical paper for measuring reliability and validity. This research is difficult to publish but must continue to be carried out in order to know the clinometric qualities of the tools. This type of studies allows to use or not to use the tools validated in clinical practice. I encourage the publication of this type of article.

Abstract is clear and well structured.

Introduction

The authors should explain why the second generation kinect might be likely to have different clinometric qualities. Better ??

This is a good comment.  I have included the following text in lines 91-96 and have also included a table summarizing the hardware differences in the Appendix (Table A1).

“For example, the K2 collects high definition RGB images (1920x1080 pixels) while the K1 collects standard definition RGB (640x480 pixels) that fails to compete with most modern webcams [25]. The RGB and IR cameras in the K2 also have wider fields of view and, when combined with updated tracking algorithms, can track greater numbers of skeletal landmarks and overall users [25].  Most importantly, the K2 utilizes a time-of-flight algorithm for motion tracking that is more robust, less noisy, and more reliable than the structured light algorithm used by the K1 [25].”

Material

Why only 5 participants ?

Thank you for this comment.  We recruited only 5 participants because it is a technical paper focused on validity and reliability with a repeatable motion.  We also felt that the large number of movements performed (total of 240 repetitions by each participant) boosted the overall ‘sample size’ in which we emphasized the number of reaches and not necessarily the number of participants.  The total sample size, when considering individual reaching motions, is really 1200 individual reaches (240 x 5 participants).

I’ve included the following text in lines 81-83:

“A small sample size was considered due to the large sample of reaches (240 repetitions) performed by each participant and the overall focus of the study being the comparison of repeatable reaching motions across sensors and testing days.”

Also in lines 308-309:

“Increasing the overall sample size in the future could mitigate these intra- and inter-individual differences in repeatable movement.”

Given the number of tests, isn't there a risk of fatigue and therefore a measurement bias?

This is an excellent comment.  Despite the large number of movements, the participants were healthy and exhibited no visible or reported signs of fatigue during testing.  They were also given short breaks between movement sets that simultaneously allowed for the researchers to save data files, name files, restart sensors, etc.

I’ve included the following text in lines 115-119:

“Given the large number of movements, participants were consistently asked for signs of fatigue and pain. None of the healthy participants reported any pain or fatigue in the UE. Participants were also given short breaks between movement sets (approx. 3-5 minutes) to mitigate fatigue. These breaks allowed researchers to code and save data files, check for data errors, and double-check or adjust experimental setup and procedures.”

Statistic is appropriate

Results

The kinematic curves could also have been analyzed with dynamic time warping (DTW) analysis

See DYSKIMOT: An ultra-low-cost inertial sensor to assess head’s rotational kinematics in adults during the Didren-Laser Test, Hage et al  in Sensors 2020

This is an advanced analysis that we did not consider.  It is an excellent suggestion.  I’ve included the following text in lines 270-273 along with the suggested reference:

“Additional, more advanced analyses may also provide further insight into these discrepancies; for example, dynamic time warping (DTW) is an advanced signal processing technique that could provide a measure of signal match for the time series data collected by the K1 and K2 [35].”

Reviewer 2 Report

Please see my comments in the notation of the attached file.

Author Response

Greetings,

I want to thank you for your thoughtful comments regarding my manuscript.  I briefly summarized your in-text comments below and have replied to each comment and highlighted changes in the new uploaded manuscript.  I have attempted to address each of your comments.  Please let me know if you have any questions and I look forward to your further review.

Thanks again,

Matthew Foreman, PhD

Reviewer 2

Thank you for your comments on the in-text references.  All references have been corrected.  There was some discrepancy with the numbering after changing reference styles. The reference page should now match the in-text references correctly, and any redundancy has been deleted.

Comment: Certain statistics tests are applicable in sample with normal distribution. As having large SD observed in Table 1, the authors need to certify the tests they used are suitable.

This is a good comment.  We did not do any normality testing with these data.  You have made this comment amongst the Bland-Altman text in line 165.  We used ANOVA w/ Bonferroni corrections, Pearson’s correlations, intraclass correlations, and Bland-Altman methods.  The data that we collected are continuous data type (measured in lengths and angles) with a large sample size (240 movements x 5 participants = 1200 total reaching movements).  Even with the large SD reported, we feel that these methods are appropriate for the large number of continuous data samples collected without normality testing.

Comment: Table 1 – the number of tests and sample size need to be indicated

I have included a description of the number of trials and sample size of each reach average both in the table description for Table 1 and within the table itself.

Comment: Figure 3 – this needs to be interpreted more for the readers

Thank you for this comment.  I have adjusted the text in lines 217-221 to read:

“The movement traces for the three planar reaching conditions (i.e., sagittal, scaption, frontal) illustrate directional differences between the Kinects and the VMC (Figure 3). Discrepancies in reaching magnitude between the Kinects and the VMC were dependent on the direction of movement. Differences in reaching ROM and planar distance were greatest during forward reaching, reduced during scaption reaching, and least during lateral reaching (Figure 3).”

Comment: Line 200 – I don’t know what this means

This statement has been removed.

Comment: Figure 3 – this statement is redundant

I have removed this statement from the description of Figure 3.

Comment: Human muscle may get tired. The markers may also move

These are good comments reflected by other reviewers as well.  The participants may have indeed gotten fatigued and the markers may have placed slightly differently between testing days.

In lines 115-119 I have mentioned the following about fatigue mitigation:

“Given the large number of movements, participants were consistently asked for signs of fatigue and pain. None of the healthy participants reported any pain or fatigue in the UE. Participants were also given short breaks between movement sets (approx. 3-5 minutes) to mitigate fatigue. These breaks allowed researchers to code and save data files, check for data errors, and double-check or adjust experimental setup and procedures.”

In lines 304-309 I have mentioned the following regarding learning effects, movement of reflective markers, and increasing sample size:

"Participants were provided verbal instruction but no formal training at the simple reaching movements, so movement may have differed between movement sets and even testing days due to subtle learning effects. It is also possible that the placement of motion capture markers varied slightly between days, resulting in reliability differences. Increasing the overall sample size in the future could mitigate these intra- and inter-individual differences in repeatable movement."

Reviewer 3 Report

The authors assessed the validity and reliability of the two generations of Microsoft Kinect for measuring trunk compensation during reaching. The paper was well written with detailed results presented. The methods were clearly explained and sound.

Main concerns:

  1. The innovation of the study was limited. The authors stated that "no previous studies have examined the abilities of both generations of the Kinect sensor for measuring trunk compensation during reaching". However, works from Valdes et al 2017 was not mentioned (Trunk Compensation During Bimanual Reaching at Different Heights by Healthy and Hemiparetic Adults). 
  2. One significance of the work was that fewer clinical tools are available given the cost and feasibility for clinical application. However, the Kinect was discontinued earlier 2018, which make it may not be a solution as well. In addition, the performance differences on Kinect sensors between healthy control and people with disabilities which were not discussed in the paper.
  3. Given the extensive experience of the authors in working with clinical populations, the paper could be much stronger with testing of a few patients with stroke instead of healthy adults.

Minor comments

background: while sufficient background was provided on using of Kinect for upper extremity movement assessment. It could be better to discuss briefly pros and cons of Kinect versus wearable sensors such as inertia sensors.

line 38: the authors could be more explicit on why directly measure the amount of compensation is important.

line 58-60: reference 22 is not for people with stroke but children with CP.

line 61-63: this statement does not seem to be supported by reference 23. The paper actually mentioned there were differences on the movement between healthy participants and people with strokes.

Methods: it was not mentioned whether training was provided for the participants and how much training. If no training, then there could be a learning effect between the two days.

Author Response

Greetings,

I want to thank you for your thoughtful comments regarding my manuscript.  I have replied to each comment below and highlighted changes in the new uploaded manuscript.  I have attempted to address each of your comments.  Please let me know if you have any questions and I look forward to your further review.

Thanks again,

Matthew Foreman, PhD

Reviewer 3

The authors assessed the validity and reliability of the two generations of Microsoft Kinect for measuring trunk compensation during reaching. The paper was well written with detailed results presented. The methods were clearly explained and sound.

Main concerns:

  1. The innovation of the study was limited. The authors stated that "no previous studies have examined the abilities of both generations of the Kinect sensor for measuring trunk compensation during reaching". However, works from Valdes et al 2017 was not mentioned (Trunk Compensation During Bimanual Reaching at Different Heights by Healthy and Hemiparetic Adults). 

Thank you for this comment.  It is true that there is plentiful literature seeking to validate the Kinect v1 and v2 for use in motion analysis.  I have tried to summarize this literature in the manuscript.  It may be true that the perceived innovation of this study is limited given the already published information comparing the Kinects to gold standard motion capture.

That being said, I believe that the innovation of this project lies in a few key areas.  First, it specifically attempts to validate the measurement of compensatory movement during reaching movements.  Other studies have looked at UE movement and posture, but few of the Kinect-specific publications have looked at simultaneous measurement of compensations during a functional task such as reaching.  Second, it seeks to validate across three sensors with the ultimate purpose of comparing the K1 to K2 to VMC.  Most existing studies compare K1 vs. VMC or K2 vs. VMC, but few compare across all three.  Finally, as mentioned in the conclusions, this study is seeking to validate and emphasize the use of the Kinect V2 moving forward in VR applications.  In reality, this is the first study in a line of work intending to (1) validate the Kinect V2 for use as a motion capture device for VR, (2) develop a VR strategy around a validated Kinect V2 for a useful and usable UE intervention for persons with stroke to perform reaching movements with compensatory movement feedback, and (3) implement this VR strategy with a small sample of persons with stroke.

I have adjusted the sentence in line 72-75 and included your reference to read:

“Few previous studies have examined the abilities of both generations of the Kinect sensor for measuring trunk compensation during reaching [24], and only one existing study has compared the measurement abilities of both sensors to simultaneous video motion capture [12].”

The Valdes study looks at reaching and trunk compensation during use with a robotic haptic device and measurement by the Kinect V1.

  1. One significance of the work was that fewer clinical tools are available given the cost and feasibility for clinical application. However, the Kinect was discontinued earlier 2018, which make it may not be a solution as well. In addition, the performance differences on Kinect sensors between healthy control and people with disabilities which were not discussed in the paper.

This is a great comment.  It is true that the Kinect V2 has been discontinued.  At the time of this work, the Kinect V2 was being used widely in rehabilitation science, particularly VR applications.  The discontinuation of the sensor was a big surprise to a lot of video gamers and researchers.  That being said, rehab science researchers are still using the Kinect V2 widely and are incorporating similar sensors that are not Microsoft branded.  We continue to use the Kinect V2 as a VR sensor for intervention and they are readily available (and even more affordable) on websites like eBay.  We have bought several recently and continue to develop with the sensor.  I think there remains a significant readership in rehab engineering, rehab science, physical therapy, occupational therapy, etc. that would be interested in this publication.

As you also mention below, the major limitation in this study is the use of healthy participants vs. stroke participants.  This is indeed a big limitation.  The purpose of the study is to validate sensors against each other and report on measurement capabilities.  Most of the referenced Kinect studies utilize healthy participants for this purpose.  However, I am aware that the tone of the manuscript is focusing on a clinical application in compensation measurement for persons with hemiparesis.

This is a notable limitation and I have attempted to address it with this paragraph starting in Line 274:

“The most notable limitation to this work is the use of healthy participants rather than a sample of participants with hemiparesis. As mentioned previously, persons with hemiparesis reach significantly differently than unimpaired persons, namely with slower movement, less accuracy, impaired interjoint coordination, and increased use of compensatory movement at the trunk [22,23]. Targets placed beyond the reach of healthy participants can illicit a similar compensatory response at the trunk, but persons with hemiparesis exhibit less symmetry and earlier trunk recruitment in comparison [23]. Healthy reaching is simply not the same as hemiparetic reaching. However, the purpose of the current study is to validate the measurement capabilities of the K1 and K2 relative to each other and to a gold standard VMC system. Numerous referenced validation studies use healthy participants for sensor validation with intentions for future clinical application [9-17]. Healthy participants are more accessible, can perform the large number of required movements without fatigue or pain, and can more readily reproduce movements across trials and testing days for validity and reliability analyses. Given that the ultimate application of this study is implementation for clinical measurement of neurologically impaired populations, the ecological validity of future work would greatly benefit from testing with a more heterogeneous sample of persons with hemiparetic stroke.”

  1. Given the extensive experience of the authors in working with clinical populations, the paper could be much stronger with testing of a few patients with stroke instead of healthy adults.

Thank you for this comment.  It is true that the study would be strengthened by the use of a small sample of stroke participants rather than healthy adults.  This was mentioned briefly in the discussion and in the final sentence (future work) in the abstract.

I have attempted to address this limitation with the paragraph described above starting in line 274.

Minor comments

background: while sufficient background was provided on using of Kinect for upper extremity movement assessment. It could be better to discuss briefly pros and cons of Kinect versus wearable sensors such as inertia sensors.

This is a good comment. There is little included text comparing the Kinects to typical motion analysis systems.  The Kinects have some advantages for implementation, specifically with low-cost VR applications.

I’ve added the following text in lines 59-64:

“The Kinect sensors have some advantages over widely used optical and inertial sensor systems, namely significantly lower cost, higher portability, easier deployment in a lab or clinic, wider accessibility, and marker-less motion tracking with simpler throughput for the control of video games and VR applications. Conversely, the Kinects typically produce significantly lower resolution and less reliable data compared to gold standard motion capture systems such as VMC and wearable inertial sensors [9,10,11,16].”

line 38: the authors could be more explicit on why directly measure the amount of compensation is important.

I have included the following sentence in lines 39-42:

“Without objective measurement and subsequent intervention, continued compensatory movements can reduce the amount of task-driven neuroplastic change achieved following neurologic injury and ultimately contribute to maladaptive plasticity, learned disuse or nonuse, and chronic pain or injury [2,3,4].”

line 58-60: reference 22 is not for people with stroke but children with CP.

This was an error in reference numbering and has been corrected.

line 61-63: this statement does not seem to be supported by reference 23. The paper actually mentioned there were differences on the movement between healthy participants and people with strokes.

This was an error in reference numbering and has been corrected.  The correct study is Levin et al. (2002) and uses reaching at different lengths to elicit trunk recruitment in healthy participants that is similar to stroke participants.  Nonetheless, stroke participants exhibit different symmetry of movement and recruit the trunk earlier in multijoint movements.

Line 69-72 now reads:

“While differences in symmetry and joint coordination exist between healthy and impaired reaching, placing objects beyond the arm’s length of healthy participants has been found to elicit trunk movement similar to that used by hemiparetic stroke patients reaching to objects within arm’s length [23].”

Methods: it was not mentioned whether training was provided for the participants and how much training. If no training, then there could be a learning effect between the two days.

This is a good comment.  Because they were healthy participants performing a simple reaching motion in different directions, there was no training other than simple verbal instruction.  Learning effects could have taken place between testing days.

I’ve included the following text in lines 111-113:

“Participants were provided verbal instruction but, given that they were healthy participants performing a relatively simple targeted reaching movement, no formal training was provided.”

And in lines 304-307:

“Participants were provided verbal instruction but no formal training at the simple reaching movements, so movement may have differed between testing days due to subtle learning effects.”

Round 2

Reviewer 2 Report

The authors have responsed most of my comments. Nonetheless, for a more clear purpose of the study, it is still suggested that the authors add some statements regarding what's the difference between the current findings and the previous one (i.e. Reither et al, 2017) they studied for upper extremity movement.

Author Response

Greetings,

Thank you for your thoughtful comments on my manuscript. I have highlighted changes within the manuscript in green to differentiate from previous edits in yellow. I have also responded to your comments below with changes and associated line numbers. Thank you for your time and effort in reviewing my manuscript, and I look forward to your thoughtful comments in the future.

Thanks again,

Matthew Foreman, PhD

Reviewer 2

The authors have responsed most of my comments. Nonetheless, for a more clear purpose of the study, it is still suggested that the authors add some statements regarding what's the difference between the current findings and the previous one (i.e. Reither et al, 2017) they studied for upper extremity movement.

Thank you for this comment!  There is a sentence addressing this in the discussion (Lines 257-260). 

I have added some additional text in the discussion section (Lines 260-267):

“In summary, Reither et al. [12] similarly found a greater range in single-day correlations between K1 and VMC (R=0.31-0.96) than between the K2 and VMC (R=0.45-0.96) with correlation magnitudes dependent on movement plane. The authors also found varied day-to-day reliability results for both K1 and K2 and, in general, a greater direction-dependent underestimation of kinematics displayed by the K1 [12]. The current study goes beyond the methods of Reither et al. [12] by utilizing an increased sample size of participants and movements, the inclusion of extended reaches to illicit trunk compensations, analysis of the trunk along with the UE, and movements in the scaption plane along with sagittal, frontal, and transverse planes.”

I have also added some additional text in the introduction, where you noted it was necessary (Lines 75-78):

“The current study aims to go beyond previous work performed in our laboratory [12] to include a larger sample size of participants and movement trials with a focus on trunk kinematics during reaches that require trunk compensation.”

Reviewer 3 Report

Thank you for addressing all the comments. The justifications for the contribution of the work and usage of healthy participants are reasonable.

Just a quick comment, as stated in the discussion session that future work with persons with hemiparetic stroke will be conducted. It would be nice to briefly discuss about what learned from this study could be applied in the future. For example, will the same 240 repetitions be needed for persons with stroke? Will the same experimental protocol be used?

Author Response

Greetings,

Thank you for your thoughtful comments on my manuscript.  I have highlighted changes within the manuscript in green to differentiate from previous edits in yellow.  I have also responded to your comments below with changes and associated line numbers.  Thank you for your time and effort in reviewing my manuscript, and I look forward to your thoughtful comments in the future.

Thanks again,

Matthew Foreman, PhD

Reviewer 3

Thank you for addressing all the comments. The justifications for the contribution of the work and usage of healthy participants are reasonable.

Just a quick comment, as stated in the discussion session that future work with persons with hemiparetic stroke will be conducted. It would be nice to briefly discuss about what learned from this study could be applied in the future. For example, will the same 240 repetitions be needed for persons with stroke? Will the same experimental protocol be used?

Thank you for this comment.  I have added an additional paragraph to describe “lessons learned” for the adjustment of future study protocols.  This paragraph is located in the discussion (Lines 300-309):

“The current study provides some insights for the design of such future work; for example, it may be necessary to recruit more individuals and reduce the overall repetitions performed to better capture variability, mitigate fatigue, and enhance the generalizability of results for real-world clinical populations. In addition, the experimental protocol could be adjusted to provide detailed instruction and training for impaired populations to reduce trial variability and enhance the efficiency of data reduction and cleaning. Given that the evidence shows that persons with hemiparetic stroke recruit the trunk earlier and more often than healthy populations [23], it may be necessary to eliminate or reduce the distance of the extended reach to maximize reaching performance and reduce frustration. Finally, given the results of the current study, it may be prudent to focus on the planes of movement best measured by the K1 and K2 due to their hardware constraints (e.g. lateral > scaption > forward).”